# Allelopathic Effect of Aqueous Extracts of Grass Genotypes on *Eruca Sativa* L.

**DOI:** 10.3390/plants12193358

**Published:** 2023-09-22

**Authors:** Masoud Motalebnejad, Hassan Karimmojeni, Mohammad Mahdi Majidi, Andrea Mastinu

**Affiliations:** 1Department of Agronomy and Plant Breeding, College of Agriculture, Isfahan University of Technology, Isfahan 84156-83111, Iran; m.mottaleb@ag.iut.ac.ir (M.M.); majidi@iut.ac.ir (M.M.M.); 2Department of Molecular and Translational Medicine, Division of Pharmacology, University of Brescia, 25123 Brescia, Italy

**Keywords:** grass genotypes, weed management, sustainability, phenolic compounds, phytocomplex

## Abstract

The aim of the current research is to evaluate the allelopathic activity of fifty grass genotypes from different species and to identify phenolic compounds in the genotypes that have the highest allelopathic activity and inhibitory effect on *Eruca sativa* L. (Rocket). Aqueous extract was prepared from the leaves of grass genotypes in different concentrations and its effect on germination and growth of *E. sativa* L. was measured. According to the results, the type of genotype and the concentration of the extract significantly decreased the percentage of germination, hypocotyl length, radicle length, and dry weight of *E. sativa* L. seedlings. Increasing the concentration of the extract resulted in a decrease in germination and growth of seedlings. The genotypes of *Festulolium* (Festulolium) (GR 5009, GR 1692, GR 5004) had the most inhibitory effect on the growth of *E. sativa* L. Also, among the genotypes studied, two genotypes (DG-M) and (DG-P) of *Dactylis glomerata* L. (orchardgrass) species showed the least allelopathic activity. The results of HPLC-MS indicated nine phenolic compounds including caffeic acid, syringic acid, vanillic acid, p-coumaric acid, ferulic acid, apigenin acid, chlorogenic acid, 4-hydroxybenzoic acid, and gallic acid. The phenolic compound most present in the aqueous extract was caffeic acid. However, phenolic compounds derived from *Festulolium* genotypes showed the greatest allelopathic action on the growth parameters of *E. sativa* L. The aqueous extracts of the *Festulolium* genotypes can be considered valid systems of sustainable weed control thanks to the phytocomplex rich in phenols.

## 1. Introduction

Weed management is an important and challenging process in the face of exploding world population, loss of available resources, and increasing environmental stress [1]. Weeds can cause yield losses of about 40%, which can reach 100% without weed control measures [2]. Weed control is essential for global food security [3] and the use of herbicides is the most common method of weed control [4]. Herbicides account for nearly half (47.5%) of the pesticides used for weed control worldwide each year [5]. The widespread and inappropriate use of synthetic herbicides leads to environmental damage (soil and water pollution), herbicide residues in food [6], and the increase of herbicide-resistant weeds [7,8], which is a major threat to food safety and human health [9].

Allelopathy can be used as a sustainable and effective strategy to prevent environmental pollution and herbicide resistance in weed control [10]. Allelopathy is a process in which one plant species stimulates or inhibits the growth of another plant species through certain secondary metabolites [11]. These secondary metabolites, called allelochemicals, enter the environment mainly through evaporation, leaching, root secretion, or decomposition of plant residues and can affect seed germination and the growth of nearby seedlings [10]. These compounds exert their effects primarily on cell division, membrane permeability, phytohormone production, photosynthesis, respiration, and enzyme activity [12]. The chemical nature of allelochemicals is complex and diverse (organic acids, aldehydes, coumarins, quinones, flavonoids, alkaloids, terpenoids, etc.), but most are produced by three main biosynthetic pathways, the shikimic acid pathway (benzoic and cinnamic acids and their derivatives, coumarins, glycosides, alkaloids, etc.) and the acetic and mevalonic acid pathways (terpenoids, steroids, complex quinones). Allelopathic compounds exist in almost all plants and are found in many plant parts such as roots, seeds, leaves, fruits, and stems [13]. Most allelochemicals are phytotoxic and interfere with the physiological parameters of target plants when they encounter the plant cell wall [14]. These phytochemicals can be used as toxic compounds to introduce new herbicides [15,16].

Phenolic compounds are among the most abundant allelochemical groups in plants and are primarily synthesized through the shikimate pathway in plants [17]. Several phenolic compounds, including vanillic acid, syringic acid, p-coumaric acid, and ferulic acid, have been identified as allelochemicals in natural and controlled ecosystems that can act as natural herbicides; therefore, plants with allelopathic activity can be used for natural weed control [18].

Allelopathy studies have been conducted on different parts of the plant and it has been found that aqueous extracts of leaves show higher germination inhibition, probably due to the higher metabolic activity of leaves, which contain more allelochemical compounds than other tissues [19,20,21]. Allelopathic aqueous extracts are water-soluble allelochemicals extracted from plants that can be used as natural herbicides and are more environmentally friendly than synthetic herbicides [15,16]. Allelopathic compounds from plant extracts have inhibitory effects on weed growth due to their diverse structures and mechanisms of action [22].

The Poaceae family contains various allelochemicals that can suppress the population and growth of weeds [23]. These allelochemicals can prevent the germination and growth of various weeds, including herbicide-resistant weeds [24]. The allelopathic effect depends on the donor and recipient species, their growth stage, and the toxicity level of the released allelochemicals [1]. The study on grass species has demonstrated the existence of phenolic compounds such as vanillic acid, chlorogenic acid, caffeic acid, ferulic acid, and coumaric acid [25] and allelopathic activity in some genotypes and species of grasses, which can have an inhibitory effect on weed germination and growth [11,26,27].

*Eruca sativa* L. is a summer annual herbaceous plant of the Brassicaceae family that is widely distributed in temperate regions [28]. *E. sativa* is widely cultivated because of its importance in industry, agriculture, and medicine [29]. However, it has become an invasive weed in some areas, causing yield losses of 6–36% in some crops, including *Sesamum indicum* and *Avena sativa* L., due to its ability to grow rapidly and high seed production. *E. sativa* seeds have a wide germination temperature range and dormancy cycle, which creates a stable seed bank in the soil and allows it to better adapt to harsh climates and low temperatures [30].

Although *Eruca sativa* L. has been known as a crop plant for a long time, it can be a weed of cool-season crops. Therefore, its biological control through allelopathy is important. For this reason, *E. sativa* L. was investigated in this study.

The present study was conducted with the aim of (i) evaluating the aqueous extract of different grass genotypes (Table 1) on the inhibition of germination and growth of *E. sativa* L., and (ii) determining and identifying phenolic compounds in the genotypes that showed the highest allelopathy activity.

## 2. Results

### 2.1. Germination Percentage

The results of the experiment show that the two factors of genotype, the concentration of plant extract, and their interaction have a significant effect (at 1% level) on the percentage of germination *Eruca sativa* L. (Table 2).

The highest decrease in the percentage of germination by (−76.5%) and (−76.13%) was observed in extract concentrations (100%) and (75%), respectively, compared to the control (Table 3), while there was no statistically significant difference at 1% level between these two treatments.

The concentration of 12.5% did not show any inhibitory effect on germination. PCA analysis (Figure 1) showed that water extracts of genotypes GR 1692, GR 5004, GR 5009, and GR 5003 of *X Festulolium* species, FA-B, FA-F, 20L-HS, 23M-HS, 10E-P, 6L-HS genotypes of *Festuca arundinacea* Schreb. (tall fescue) species, LP-AR1 genotype of *Lolium perenne* L. (perennial ryegrass) species, LM-AL genotype of *Lolium multiflorum* Lam. (Italian ryegrass) species, BI-G25 genotype of *Bromus inermis* L. (smooth bromegrass) species, and LH-T genotype of *Lolium×hybridum* Hausskn species had the greatest inhibitory effect on *E. sativa* L. and caused a decrease in germination up to 46.1%. Among these genotypes, *Festulolium* genotypes showed the highest growth inhibition effect (Table 4).

### 2.2. Hypocotyl Length 

The results showed that the factor of genotype type, extract concentration (12.5, 25, 50, 75, 100%), and also the interaction effect of these two factors had a significant effect (at 1% level) on the length of the hypocotyl of *E. sativa* L. (Table 2). The greatest decrease in hypocotyl length was observed by (79.36–78.86%) in the concentrations of (100%) and (75%), respectively, compared to the control (Table 3). Also, the concentration of 12.5% was not different from the control and showed no inhibitory effect on hypocotyl length. PCA analysis showed that water extracts from genotypes GR 1692, GR 5004, GR 5009, GR 5003 of *X Festulolium* species, FA-B, FA-F, 20L-HS, 23M-HS, 10E-P, 6L-HS genotypes of *F. arundinacea* Schreb. species, LP-AR1 genotype of *L. perenne* L. species, LM-AL genotype of *L. multiflorum* Lam. species, BI-G25 genotype of *B. inermis* Leyss species, and LH-T genotype of *Lolium × hybridum* Hausskn species had the greatest inhibitory effect on *E. sativa* L. and caused a decrease in hypocotyl length up to 43.9%. Among these genotypes, *Festulolium* genotypes showed the highest growth inhibition effect (Figure 1, Table 4).

### 2.3. Radicle Length

The results of our study showed that the effect of factors such as genotype, concentration of plant extract (12.5, 25, 50, 75, 100%), and their interaction on radicle length of *E. sativa* L. is significant at 1% level (Table 2). Application of the extract at the highest concentration caused the greatest reduction in radicle length by 81.4% compared to the control. Application of the extract at the lowest concentration (12.5%) did not show any inhibitory effect on radicle length (Table 3). PCA analysis showed that water extracts from genotypes GR 1692, GR 5004, GR 5009, GR 5003 of *X Festulolium* species, genotypes FA-B, FA-F, 20L-HS, 23M-HS, 10E- P, 6L-HS of *F. arundinacea* Schreb. Species, LP-AR1 genotype of *L. perenne* L. species, LM-AL genotype of *L. multiflorum* Lam. Species, BI-G25 genotype of *B. inermis* Leyss species, and LH-T genotype of *Lolium×hybridum* Hausskn species had the greatest inhibitory effect on radicle length and reduced radicle length by 50.7%. Among these genotypes, *Festulolium* genotypes showed the highest growth inhibition effect (Figure 1, Table 4).

### 2.4. Seedling Dry Weight

The results of the study showed that genotype factors, extract concentration (12.5, 25, 50, 75, 100%), and the interaction of these two factors had a significant effect (at 1% level) on the dry weight of *E. sativa* L. seedlings (Table 2). The maximum decrease in dry weight of seedlings was observed (−75.63%) in the presence of the highest extract concentration (100%). The 12.5% extract concentration did not show any inhibitory effect on seedling dry weight (Table 3). PCA analysis showed that water extracts of genotypes GR 1692, GR 5004, GR 5009, GR 5003 of *X Festulolium* species, FA-B, FA-F, 20L-HS, 23M-HS, 10E-P, 6L-HS genotypes of *F. arundinacea* Schreb. species, LP-AR1 genotype of *L. perenne* L. species, LM-AL genotype of *L. multiflorum* Lam. species, BI-G25 genotype of *B. inermis* Leyss species, and LH-T genotype of *Lolium×hybridum* Hausskn species had the greatest inhibitory effect on seedling dry weight of *E. sativa* L., reducing dry weight by 61%. Among these genotypes, *Festulolium* genotypes showed the highest growth inhibition effect (Figure 1, Table 4).

### 2.5. Total Phenol and Flavonoid

Among the fifty genotypes studied, three genotypes with the highest allelopathic activity and two genotypes with the lowest allelopathic activity were selected according to the results of the comparison of means and PCA for the study of phenolic and flavonoid compounds. The results showed the presence of phenolic and flavonoid compounds in the shoots of the grass genotypes (Table 5). The results of the comparison of means showed that genotypes *X Festulolium* sp. (GR 5009), *X Festulolium* sp. (GR 1692), and *X Festulolium braunii* (K. Richt.) (GR 5004) have more total phenolics and flavonoids (Table 5) and it seems that the high allelopathic activity of these genotypes could be due to their high phenolic compounds.

### 2.6. High-Performance Liquid Chromatography-MS (HPLC-MS) Analysis

The HPLC-MS results indicate the presence of phenolic compounds in the shoots of the grass genotypes (Table 6). Nine phenolic compounds were identified in the shoots of the grass genotypes, including caffeic acid, syringic acid, vanillic acid, p-coumaric acid, ferulic acid, apigenin acid, chlorogenic acid, 4-hydroxybenzoic acid, and gallic acid (Table 6). The HPLC-MS results showed that the shoot parts of the genotypes of *Festulolium* species have the highest amount of phenolic compounds. Among the nine identified phenolic compounds, the highest amount was related to caffeic acid, and it seems to be one of the most important phenolic compounds effective in allelopathy (Table 6). According to the HPLC-MS results, these compounds seemed to have the greatest effect in inhibiting the growth of *E. sativa*.

## 3. Discussion

The results of our studies showed that the aqueous extract of the leaves of the grass genotypes decreased the percentage of germination, hypocotyl length, radicle length, and dry weight of *Eruca sativa* L. seedlings. It is also known that the amount and variety of allelochemicals in the leaves of grasses are high, which may have an inhibitory effect on the germination and growth of other plants [31]. The study conducted by Shi et al. (2023) showed that the aqueous extract of *Abutilon theophrasti* leaves at high concentrations had an inhibitory effect on the germination and growth of *Glycine max* L., *Triticum aestivum* L., *Zea mays* [32]. In another study on the allelopathic effects of aqueous leaf extracts of two grass species, *Urochloa decumbens* and *Urochloa ruziziensis*, it was shown that they had inhibitory effects on the weeds *Chloris ventricosa*, *Bidens pilosa*, *Commelina benghalensis* L., *Conyza canadensis*, and *Digitaria insularis* and caused a decrease in the percentage of germination, root growth, shoot growth, and biomass of the weeds [33].

It was determined in our study that as the concentration of the extract increased, a further decrease in the traits studied was observed (Table 2 and Table 3). The results of our study are consistent with previous studies that found that increasing the concentration of allelopathic plant extracts further reduced the germination and growth of target plants. In a study, the allelopathic effects of aqueous leaf extract of *Cannabis sativa* at concentrations of 25, 50, 75, and 100% on seed germination and seedling growth of *Triticum durum* and *Hordeum vulgare* were investigated, and it was found that the allelopathic effect of the extract was concentration-dependent and germination decreased with increasing concentration of the extract [34]. Hussain et al. (2020) investigated the allelopathic effects of *Acacia melanoxylon* R. Br shoot aqueous extract under laboratory conditions and at concentrations (0, 25%, 50%, 75%, and 100%) on the growth of *Lactuca sativa* seedlings and reported that germination, shoot length, root length, and dry weight of seedlings decreased after exposure to *A. melanoxylon* aerial extract, and the greatest decrease was observed at 75% and 100% concentrations. The allelopathic effects of *A. melanoxylon* extract may be due to the presence of phenolic and flavonoid allelochemical compounds, which often have inhibitory effects on the growth of target species [15].

Seed germination and the early stages of seedling growth are the most sensitive stages to environmental changes, so this stage is often used to study allelopathic effects [35,36]. It appears that the reduction in seed germination at higher concentrations of aqueous leaf extracts occurs because of reduced water uptake by seeds due to the presence of allelochemicals in the absorptive substrate. Seed uptake of allelochemicals leads to seed toxicity, which ultimately severely reduces water and nutrient uptake and arrests seedling growth and development [37]. Changes in gibberellic hormone activity, which regulates amylase production during germination, can also occur in the presence of allelochemicals (phenolic compounds) [38]. Susceptibility to allelopathic compounds may depend on small seed size. Small seeds and early-emerging species have been reported to be more susceptible to allelopathic effects than plants with larger seeds. Small seeds are more susceptible to allelopathy because of reduced carbohydrate storage [39].

Our study showed that the reduction in growth of *E. sativa* L. seedlings caused by the aqueous extract was observed more in the rootlets than in the shoot. Therefore, the rootlet is more sensitive to allelochemicals than the shoot, which may be because the rootlets are in direct contact with allelochemical compounds [36]. Also, the permeability of root tissue to allelopathic compounds is higher than that of the shoot, which makes the rootlet more sensitive to these compounds [40]. The reduction in root growth under the influence of allelopathic compounds may be due to the disruption of mitosis, resulting in a decrease in root length and a concomitant decrease in root volume. This effect on root growth may be responsible for the decrease in germination, shoot length, root length, and seedling dry weight due to the decrease in moisture and nutrient uptake [1]. Therefore, seedling growth, especially root growth, can be considered as a good indicator of plant sensitivity to allelopathy [41].

The reason for the decrease in dry weight is allelochemical toxicity, which causes a decrease in water uptake in tissues [42]. Allelochemical stress increases the concentration of reactive oxygen species (ROS) in plant cells [43]. As a result, ROS cause oxidative damage and increase lipid peroxidation in the membrane. Lipid peroxidation causes changes in the fluidity and permeability of lipid bilayer membranes, which can alter cell integrity and ultimately lead to cell death [44]. Allelopathic compounds can also reduce the activity of metabolic enzymes, proteins, carbohydrates, and nucleic acid content. This decrease in metabolite and enzyme activity is considered the mechanism of action of allelochemicals and provides the basis for further studies on the use of allelopathic plant extracts as biological herbicides for weed control [19].

Our study showed that different grass genotypes have different inhibitory effects on the growth of *E. sativa* L. (Figure 1, Table 4), such that the highest growth reduction was caused by the genotypes of *Festulolium* species including GR 1692, GR 5004, and GR 5009 (Figure 1, Table 4). The different allelopathic effects of plants on the target species may be due to the presence of different levels of allelochemicals in plants, which is caused by the different abilities of plants to synthesize allelopathic substances [32,45]. Previous studies have shown that there is variation among different grass species in their ability to suppress weeds, and the allelopathic effect varies among different species and genotypes. Lipinska et al. (2019) investigated the allelopathic potential of six cultivars of *Festuca arundinacea*, *Festuca ovina*, and reported that *Festuca rubra* had different inhibitory effects on the growth of grass weeds, and their allelopathic potential depended on the content of flavonoids and phenolic acids in their leaves [46]. In the study of Koo et al. (2022), the aqueous extract of the leaves of different cultivars of *Lolium arundinaceum* showed an inhibitory effect on the germination and growth of *Poa annua* L. in a petri dish, and there was a difference between different cultivars of *L. arundinaceum* in terms of allelopathic effects [47].

Advances in compound isolation techniques allow the identification of active compounds in allelopathy [48]. The results obtained from our study showed that genotypes of *Festulolium* species have high levels of phenolic compounds, which can have an inhibitory effect on the growth of the target species (Table 6). Based on HPLC-MS data, it was found that the shoot parts of *Festulolium* species genotypes (GR 5009, GR 1692) have phenolic compounds including caffeic acid, syringic acid, vanillic acid, p-coumaric acid, ferulic acid, apigenin acid, chlorogenic acid, 4-hydroxybenzoic acid, and gallic acid (Table 6). These nine compounds had the highest concentration, and it seems that they were effective compounds in the allelopathic activity of *Festulolium* species genotypes and caused an inhibitory effect on the germination and growth of *E. sativa* L. (Table 6). Phenolic compounds are among the most important secondary metabolites involved in allelopathy. These compounds can increase lipid peroxidation, ultimately leading to reduced growth or death of plant tissue [49]. In addition, phenolic allelochemicals prevent cell elongation and division by reducing nutrient uptake by plants, causing changes in plant cell structure, and reducing plant growth [18]. In other studies, the presence of phenolic compounds in allelochemicals has been proven. The presence of phenolic compounds in *Helianthus annuus* was shown and it was found that ferulic acid has the highest amount followed by vanillic acid, chlorogenic acid, and caffeic acid [50]. The aqueous extract of *Lolium perenne* L. leaves contains phenolic compounds, the highest amount of which is related to chlorogenic acid [51]. Allelopathic phenolic compounds of caffeic acid, p-coumaric acid, and ferulic acid in *Lolium multiflorum* reduced shoot and root length in rice cultivars [52]. These allelochemicals can disrupt plant physiological and biological processes, reducing or suppressing plant growth and development by reducing mineral uptake by the plant [53,54].

## 4. Materials and Methods

### 4.1. Plant Materials and Methods of Preparation

The grass genotypes (Table 1) at the research farm of Isfahan University of Technology in Lavark, Najaf Abad, Iran (40 km southwest of Isfahan, 32°32′, N 51°, 23′ E and 1630 m above sea level) were cultivated and used in 2015 [55]. Fresh samples of 50 grass genotypes from ten species were collected in late spring and at the flowering stage in 2019 and placed separately in paper bags. The samples were air-dried at room temperature, then powdered and stored in closed plastic bags at room temperature until use. The seeds of *E. sativa* L. were obtained from Pakan Bazr Co., Isfahan, Iran, and their germination percentage and dormancy were tested in petri dish conditions in the germinator.

### 4.2. Preparation of Leaf Extract of Grass Genotypes and Germination Experiments

Leaves of grass genotypes harvested at the flowering stage were used to prepare the extract. Samples were shade-dried. The method of Bali et al. (2016) was used to prepare the aqueous leaf extract [56]. For the extraction, the samples collected from the farm were first powdered and 12.5, 25, 50, 75, and 100 g of the powdered samples of each treatment were poured into 100 mL of deionized water and kept for 24 h at a temperature of 25 °C. Then, the prepared extracts were passed through Whatman #1 filter paper, and the extract of the prepared samples was stored at 4 °C until use. The seeds of *E. sativa* L. were first disinfected with 10% sodium hypochlorite solution for 10 min, then washed with normal water for 10 min and finally with deionized water for 5 min.

Fifty seeds of *E. sativa* L. were placed on filter paper in 9 cm petri dishes. The seeds were soaked with the prepared extracts (10 mL for each petri dish) and the petri dishes were placed in the growth chamber at 25°C and 12 h of light. Irrigation with distilled water was used as a control. After seed germination was determined, the percentage of germination and seedling growth were examined [34]. To calculate the dry weight of seedlings, samples were dried in an oven at 70 °C for 48 h and then weighed [31].

### 4.3. Measurement of Total Phenolic and Flavonoid Content in Leaves of Grass Genotypes

For this purpose, based on the results of comparison of means and PCA, three grass genotypes with the highest and two genotypes with the lowest allelopathic activity were selected among the genotypes for the measurement of total phenolics and flavonoids in them. The total concentrations of phenolics and flavonoids in the plant shoots were estimated using gallic acid as a standard and the Folin–Ciocalteu colorimetric method [57]. Results were expressed as gallic acid equivalents (GAE).

Briefly, 3 g of air-dried sample (leaf powder samples) was extracted with 10 mL of 80% methanol using an orbital shaker incubator (Jaltajhiz, Iran, Karaj, JTSL20) (110 rpm) at 25 °C for 24 h. A 0.5 mL aliquot of the methanol extract was then filtered and combined with 2.5 mL of Folin–Ciocalteu reagent (diluted with 1:10 volume of distilled water) and 2 mL of 7.5% (*v*/*v*) sodium carbonate. It was heated at 45 °C for 15 min and the absorbance was measured at 765 nm against a blank by spectrophotometry (HITAGHI-Japan model U-1800). The phenolic content of the shoots was recorded as gallic acid equivalents per 1 g of shoot dry matter.

### 4.4. Identification of Phenolic Compounds

For this purpose, five genotypes (three genotypes with high allelopathic activity and two genotypes with low allelopathic activity), whose total phenolics and flavonoids were studied in the previous step, were used. Leaf powder samples were extracted with 80% methanol [58]. A 100 g sample of each leaf powder was extracted with 300 mL of 80% methanol (HPLC grade, Merck) (stirring, 25 °C for 48 h, centrifugation, 1200× *g* for 15 min). The extracts were analyzed on an HPLC-MS system (model Agilent 1090). The instrument consisted of an Agilent 1100 HPLC, diode detector and mass spectrometer (MSD, SL mode) (Agilent Technologies, Palo Alto, CA, USA). The extracts were filtered through a 0.22 μm Acrodisc nylon filter. Injections on the analytical column were made from 20 μL of filtered extract. Standards were dissolved using HPLC grade methanol as the solvent. The stationary phase consisted of a 250 mm × 4.6 mm (5 μm) symmetrical C18 column (Waters Crop., Milford, MA, USA) (10 mm × 4 mm ID), and the mobile phase was formic acid (0.1%). Acetonitrile (99.8%) was used at a flow rate of 0.8 mL/min and the wavelength was set between 200 and 400 nm. The implementation of the gradient conditions was characterized by the following specifications: 10–26% solvent B (*v*/*v*) for 40 min, 65% solvent B for 70 min, and finally 100% solvent B for 75 min. Phenolic content was determined by setting the DAD at 350, 310, 270, and 520 nm, reading extreme peaks in real time, and continuously recording the entire spectrum (190–650 nm). Analysis of allelopathic compounds was repeated three times using extracts from each sample.

### 4.5. Statistical Analysis

The experiment was conducted as a completely randomized factorial design with three replications, and all statistical calculations were performed using SAS 9.4. The Kolmogorov–Smirnov test was used to check the normality of the data distribution before performing the analysis of variance (ANOVA). Principal component analysis (PCA) was performed on the correlation matrix of the traits using XLSTAT software version 2019.2.2.

## 5. Conclusions

According to this study, aqueous extracts of grass genotypes with concentration of 100% and 75% have significant allelopathic activity on germination and growth of *E. sativa* L. There was a difference between different grass genotypes in terms of allelopathic activity, so the studied genotypes were classified into two weak and strong groups in terms of inhibition of *E. sativa* L. growth. The genotypes of *Festulolium* including GR 1692, GR 5004, GR 5009 had the most inhibitory effect on the growth of *E. sativa*. It can be concluded from the present study that it is possible to use the extract of the tested genotypes as a natural alternative method for the control of *E. sativa* weeds. From the present study, it can be concluded that the tested genotypes of *Festulolium* can be used as a natural alternative method to control *E. sativa* weeds. Finally, it is suggested that further studies should be conducted to identify effective allelochemicals in the allelopathy of these genotypes.

## Figures and Tables

**Figure 1 plants-12-03358-f001:**
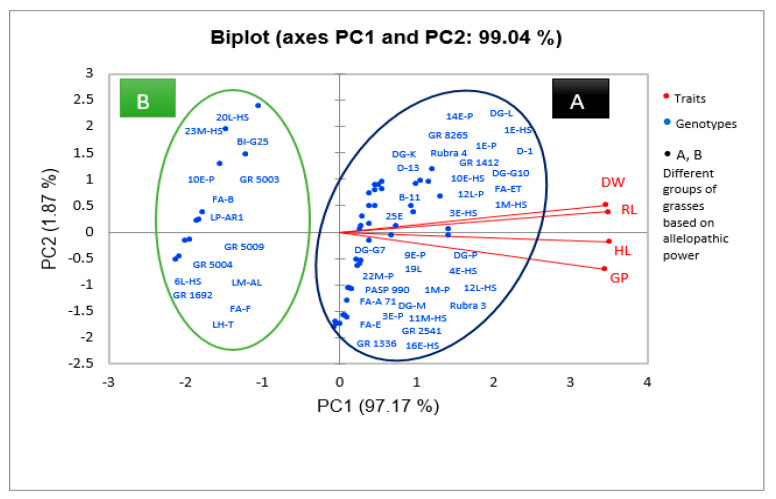
Principal component analysis (PCA) of extracts of 50 grass genotypes of the family Poaceae with respect to their germination indices. According to the results obtained, two components explained 99% of the total variation. Therefore, the genotypes were divided into two groups. Group A comprises the genotypes that had the least inhibitory effect on the traits studied and showed high values of germination, rootlet length, shootlet length, and dry weight. Group B includes the genotypes that had the greatest inhibitory effect on the traits studied and caused a significant decrease in germination traits, indicating their greater allelopathic activity.

**Table 1 plants-12-03358-t001:** Information on 50 grass genotypes used for aqueous extract.

Origin	Variety	Genotype Code	Specie
-	-	FA-ET	*Festuca arundinacea* Schreb.
Germany, Zurich	Belfine	FA-B	*Festuca arundinacea* Schreb.
Germany, Zurich	Elfina	FA-E	*Festuca arundinacea* Schreb.
Croatia	B-18	GR 1336	*Festuca arundinacea* Schreb.
New Zealand	Roa	GR 1412	*Festuca arundinacea* Schreb.
Iran, Isfahan, Fozve	16Early-Half Sib	16E-HS	*Festuca arundinacea* Schreb.
Romania	Cluj	GR 8265	*Festuca arundinacea* Schreb.
Spain	-	FA-A	*Festuca arundinacea* Schreb.
Iran, Isfahan, Fozve	9Early-Parent	9E-P	*Festuca arundinacea* Schreb.
Hungary, unknown	11Moderate-Half Sib	11M-HS	*Festuca arundinacea* Schreb.
Iran, Kohkiluye, Yasuj	6Late-Half Sib	6L-HS	*Festuca arundinacea* Schreb.
Poland, unknown	22Moderate-Parent	22M-P	*Festuca arundinacea* Schreb.
USA, New Jersey	10Early-Half Sib	10E-HS	*Festuca arundinacea* Schreb.
Hungary, unknown	14Early-Parent	14E-P	*Festuca arundinacea* Schreb.
Iran, Isfahan, Fozve	20Late-Half Sib	20L-HS	*Festuca arundinacea* Schreb.
Hungary, unknown	12Late-Half Sib	12L-HS	*Festuca arundinacea* Schreb.
Iran, Isfahan, Yazdabad	1Early-Parent	1E-P	*Festuca arundinacea* Schreb.
Iran, Shahrud	-	19L	*Festuca arundinacea* Schreb.
Iran, Isfahan, Yazdabad	1Moderate -Parent	1M-P	*Festuca arundinacea* Schreb.
Iran, Kohkiluye, Yasuj	3Early-Half Sib	3E-HS	*Festuca arundinacea* Schreb.
Iran, Isfahan, Yazdabad	1Moderate-Half Sib	1M-HS	*Festuca arundinacea* Schreb.
Iran, Shahrud	-	25E	*Festuca arundinacea* Schreb.
Iran, Kohkiluye, Yasuj	3Early-Parent	3E-P	*Festuca arundinacea* Schreb.
Iran, Isfahan, Mobarake	4Early-Half Sib	4E-HS	*Festuca arundinacea* Schreb.
USA, New Jersey	10Early-Parent	10E-P	*Festuca arundinacea* Schreb.
Hungary, unknown	12Late-Parent	12L-P	*Festuca arundinacea* Schreb.
Iran, Isfahan, Yazdabad	1Early-Half Sib	1E-HS	*Festuca arundinacea* Schreb.
Poland, unknown	23Moderate-Half Sib	23M-HS	*Festuca arundinacea* Schreb.
France	Flecha	FA-F	*Festuca arundinacea* Schreb.
Iran, Isfahan, IUT	-	Rubra 4	*Festuca rubra* L.
Iran, Isfahan, IUT	-	Rubra 3	*Festuca rubra* L.
Hungary	-	BI-G25	*Bromus inermis* Leyss.
Hungary	RCAT042134	B-11	*Bromus inermis* Leyss.
France	Medly	DG-M	*Dactylis glomerata* L.
Germany, Zurich	PRATO	DG-P	*Dactylis glomerata* L.
Hungary	RCAT041111	DG-G;7	*Dactylis glomerata* L.
France	Ludac	DG-L	*Dactylis glomerata* L.
Iran, Isfahan, Fozve	4000 ∕ 24	DG-G10	*Dactylis glomerata* L.
Iran, Isfahan, Najaf Abad	31.4000	D-1	*Dactylis glomerata* L.
France	Kasbah	DG-K	*Dactylis glomerata* L.
Hungary	RCAT041111	D-13	*Dactylis glomerata* L.
Netherland	Baroldi, Barwoldi, Barenza	GR 2541	*Lolium multiflorum* Lam.
Germany, Zurich	Alces	LM-AL	*Lolium multiflorum* Lam.
Germany, Zurich	Arvella	LP-AR1	*Lolium perenne* L.
Germany, Zurich	Tapirus	LH-T	*Lolium × hybridum* Hausskn.
CSFR	Perun	GR 5004	*X Festulolium braunii* (K. Richt.)
Germany	Paulita	GR 5003	*X Festulolium braunii* (K. Richt.)
Germany	F1 3.79	GR 5009	*X Festulolium* sp.
Netherland	Civ 254	GR 1692	*X Festulolium* sp.
Spain	-	PASP 9;90	*Paspalum dilatatum* Poir.

**Table 2 plants-12-03358-t002:** Analysis of variance of the effect of the grass genotypes and extract concentrations on the germination indices of *Eruca sativa* L.

Source of Variation	df	Mean Square			
		Germination	Hypocotyl Length	Radicle Length	Dry Weight
Extract concentration	5	148,872 **	31,270 **	4954 **	845 **
Genotypes	49	398 **	287 **	38.6 **	28.9 **
Genotypes × Extract concentration	245	99.6 **	59.6 **	3.87 **	0.78 **
Error	600	1.58	0.61	0.145	0.12
Coefficient of variation		15.7	13.9	12.3	17.4

** Significant at 1% level of probability.

**Table 3 plants-12-03358-t003:** Mean comparison for seed germination indices of *Eruca sativa* L. affected by the extract concentrations of the grass genotypes.

Extract Concentration	Germination (%)	Hypocotyl Length (mm)	Radicle Length (mm)	Dry Weight (mg Plant)
Control	90 ± 1.08 ^a^	49.44 ± 0.43 ^a^	19.25 ± 0.37 ^a^	8.29 ± 0.15 ^a^
12.5	90.13 ± 0.80 ^a^	49.62 ± 0.59 ^a^	19.22 ± 0.49 ^a^	8.27 ± 0.09 ^a^
25	71.28 ± 0.92 ^b^	42.49 ± 1.42 ^b^	15.51 ± 0.34 ^b^	7.05 ± 0.10 ^b^
50	50.56 ± 0.55 ^c^	36.90 ± 0.85 ^c^	13.02 ± 0.52 ^c^	5.76 ± 0.17 ^c^
75	21.48 ± 1.35 ^d^	10.39 ± 1.27 ^d^	7.21 ± 0.19 ^d^	3.15 ± 0.12 ^d^
100	21.16 ± 1.14 ^d^	10.20 ± 1.06 ^d^	3.58 ± 0.21 ^e^	2.02 ± 0.06 ^e^
LSD (5%)	0.349	0.237	0.081	0.074

Means followed by the same letter in each column are not significantly different according to the LSD test at the 5% level of probability.

**Table 4 plants-12-03358-t004:** Mean comparison of the seed germination indices of *E. sativa* affected by the effects of the 50 grass genotypes.

Grass Genotyp	Germination (%)	Hypocotyl Length (mm)	Radicle Length (mm)	Dry Weight(mg)
*Festuca arundinacea* Schreb. (FA-ET)	63.22 ± 6.09 ^b^	40.41 ± 4.17 ^ab^	14.67 ± 1.82 ^ab^	7.79 ± 0.39 ^ab^
*Festuca arundinacea* Schreb. (FA-B)	49.27 ± 7.36 ^lmn^	28.39 ± 4.09 ^t–w^	10.01 ± 1.56 ^yz^	3.89 ± 0.56 ^r^
*Festuca arundinacea* Schreb. (FA-E)	59.38 ± 6.19 ^fg^	36.62 ± 3.39 ^m–q^	12.77 ± 1.39 ^u–y^	5.64 ± 0.67 ^mno^
*Festuca arundinacea* Schreb. (GR 1336)	59.22 ± 5.82 ^fgh^	36.58 ± 3.17 ^n–q^	12.72 ± 1.09 ^u–y^	5.63 ± 0.38 ^mno^
*Festuca arundinacea* Schreb. (GR 1412)	58.55 ± 4.38 ^h–k^	37.69 ± 4.21 ^fgh^	13.46 ± 1.14 ^m–q^	6.39 ± 0.29 ^hi^
*Festuca arundinacea* Schreb. (16E-HS)	59.11 ± 5.44 ^f–j^	36.48 ± 4.32 ^o–r^	12.66 ± 1.12 ^wxy^	5.62 ± 0.51 ^mno^
*Festuca arundinacea* Schreb. (GR 8265)	59.08 ± 5.12 ^f–j^	37.96 ± 4.25 ^fg^	13.8 ± 1.33 ^g–j^	6.69 ± 0.37 ^fg^
*Festuca arundinacea* Schreb. (FA-A 71)	58.83 ± 7.04 ^f–k^	36.91 ± 3.01 ^k–o^	12.91 ± 1.02 ^t–x^	5.74 ± 0.39 ^m^
*Festuca arundinacea* Schreb. (9E-P)	61 ± 9.23 ^e^	37.94 ± 4.30 ^fg^	13.89 ± 1.21 ^fghi^	6.7 ± 0.36 ^fg^
*Festuca arundinacea* Schreb. (11M-HS)	59.18 ± 4.87 ^fgh^	36.3 ± 3.44 ^pqr^	12.62 ± 2.22 ^xy^	5.55 ± 0.40 ^mno^
*Festuca arundinacea* Schreb. (6L-HS)	48.94 ± 6.36 ^lmno^	28.07 ± 5.21 ^vwx^	9.86 ± 1.30 ^yz^	3.56 ± 0.38 ^s^
*Festuca arundinacea* Schreb. (22M-P)	59.13 ± 5.02 ^f–j^	37.18 ± 3.42 ^i–l^	13.17 ± 1.44 ^r–v^	6.06 ± 0.40 ^kl^
*Festuca arundinacea* Schreb. (10E-HS)	61.2 ± 5.91 ^de^	39.61 ± 3.61 ^de^	14.21 ± 1.19 ^de^	7.4 ± 0.42 ^de^
*Festuca arundinacea* Schreb. (14E-P)	58.78 ± 7.49 ^f–k^	37.90 ± 3.37 ^fg^	13.7 ± 1.25 ^h–l^	6.61 ± 0.37 ^gh^
*Festuca arundinacea* Schreb. (20L-HS)	49.67 ± 9.05 ^l^	32.37 ± 7.41 ^s^	11.58 ± 1.36 ^xyz^	4.89 ± 0.81 ^p^
*Festuca arundinacea* Schreb. (12L-HS)	58.88 ± 3.93 ^f–j^	37.57 ± 4.22 ^f–i^	13.41 ± 1.41 ^m–r^	6.31 ± 0.40 ^ij^
*Festuca arundinacea* Schreb. (1E-P)	61.5 ± 5.32 ^cd^	39.85 ± 4.07 ^cde^	14.31 ± 1.56 ^cd^	7.51 ± 0.39 ^cd^
*Festuca arundinacea* Schreb. (19L)	58.80 ± 6.07 ^f–k^	37.13 ± 3.59 ^i–m^	13.02 ± 1.27 ^t–x^	6.03 ± 0.42 ^l^
*Festuca arundinacea* Schreb. (1M-P)	58.94 ± 6.23 ^f–j^	37.02 ± 3.27 ^j–n^	12.96 ± 1.57 ^t–x^	5.75 ± 0.37 ^m^
*Festuca arundinacea* Schreb. (3E-HS)	61.3 ± 6.14 ^cde^	39.55 ± 3.09 ^e^	14.10 ± 1.38 ^def^	7.22 ± 0.29 ^e^
*Festuca arundinacea* Schreb. (1M-HS)	61.2 ± 5.79 ^de^	40.16 ± 4.19 ^bc^	14.02 ± 1.82 ^efg^	7.23 ± 0.29 ^e^
*Festuca arundinacea* Schreb. (25E)	58.33 ± 6.41 ^jk^	37.24 ± 4.83 ^h–l^	13.22 ± 1.30 ^q–u^	6.11 ± 0.28 ^jkl^
*Festuca arundinacea* Schreb. (3E-P)	59 ± 6.55 ^f^	36.77 ± 4.15 ^l–p^	12.81 ± 1.16 ^u–y^	5.7 ± 0.26 ^mn^
*Festuca arundinacea* Schreb. (4E-HS)	59.16 ± 6.61 ^f–j^	37.13 ± 4.33 ^ijkl^	13.11 ± 1.42 ^s–w^	6.06 ± 0.33 ^kl^
*Festuca arundinacea* Schreb. (10E-P)	49.38 ± 5.69 ^lm^	28.51 ± 6.20 ^tu^	10.06 ± 1.45 ^yz^	4 ± 0.75 ^r^
*Festuca arundinacea* Schreb. (12L-P)	59.16 ± 6.04 ^f–j^	37.94 ± 3.49 ^fg^	13.76 ± 1.37 ^h–k^	6.66 ± 0.37 ^fg^
*Festuca arundinacea* Schreb. (1E-HS)	58.61 ± 5.87 ^g–k^	37.86 ± 3.32 ^fg^	13.65 ± 1.20 i^–m^	6.56 ± 0.45 ^gh^
*Festuca arundinacea* Schreb. (23m-HS)	49.61 ± 4.84 ^l^	28.72 ± 5.75 ^t^	11.43 ± 1.16 ^yz^	4.25 ± 0.59 ^q^
*Festuca arundinacea* Schreb. (FA-F)	58.72 ± 5.92 ^f–k^	36.05 ± 4.11 ^r^	12.45 ± 1.52 ^xyz^	5.42 ± 0.39 ^o^
*Festuca rubra* L. (Rubra 4)	58.16 ± 6.40 ^k^	37.86 ± 3.66 ^fg^	13.49 ± 2.07 ^l–p^	6.39 ± 0.41 ^hi^
*Festuca rubra* L. (Rubra 3)	61.16 ± 6.17 ^de^	38.04 ± 3.60 ^f^	13.94 ± 1.49 ^fgh^	6.84 ± 0.38 ^f^
*Bromus inermis* Leyss. (BI-G25)	49.65 ± 7.08 ^l^	32.19 ± 4.05 ^s^	11.48 ± 1.24 ^yz^	4.31 ± 0.53 ^q^
*Bromus inermis* Leyss. (B-11)	58.44 ± 7.43 ^ijk^	37.29 ± 3.42 ^h–k^	13.27 ± 1.51 ^p–t^	6.17 ± 0.31 ^jkl^
*Dactylis glomerata* L. (DG-M)	64.55 ± 6.86 ^a^	40.47 ± 3.30 ^ab^	14.7 ± 1.32 ^ab^	7.88 ± 0.38 ^a^
*Dactylis glomerata* L.(DG-P)	64.33 ± 6.23 ^a^	40.91 ± 3.69 ^a^	14.83 ± 1.46 ^a^	7.68 ± 0.36 ^abc^
*Dactylis glomerata* L. (DG-G7)	59.27 ± 6.18 ^fg^	37.52 ± 4.31 ^ghi^	13.37 ± 1.05 ^n–r^	6.27 ± 0.29 ^ijk^
*Dactylis glomerata* L. (DG-L)	58.66 ± 4.57 ^g–k^	37.86 ± 4.54 ^fg^	13.59 ± 2.31 ^j–n^	6.56 ± 0.36 ^gh^
*Dactylis glomerata* L. (DG-G10)	61.94 ± 5.96 ^cd^	40.38 ± 5.11 ^b^	14.46 ± 1.71 ^bc^	7.62 ± 0.36 ^bcd^
*Dactylis glomerata* L. (D-1)	62.11 ± 6.09 ^c^	40.11 ± 4.40 ^bcd^	14.57 ± 1.53 ^b^	7.74 ± 0.38 ^ab^
*Dactylis glomerata* L. (DG-K)	58.34 ± 7.33 ^jk^	37.32 ± 3.24 ^h–k^	13.33 ± 1.17 ^o–s^	6.22 ± 0.35 ^i–l^
*Dactylis glomerata* L. (D-13)	59.07 ± 6.26 ^f–j^	37.83 ± 5.16 ^fg^	13.55 ± 1.79 ^k–o^	6.53 ± 0.36 ^gh^
*Lolium multiflorum* Lam. (GR 2541)	58.74 ± 7.18 ^f–k^	36.81 ± 3.23 ^k–p^	12.85 ± 1.35 ^t–x^	5.74 ± 0.38 ^m^
*Lolium multiflorum* Lam. (LM-AL)	58.77 ± 7.01 ^f–k^	36.27 ± 4.38 ^qr^	12.5 ± 1.31 ^xyz^	5.49 ± 0.39 ^o^
*Lolium perenne* L. (LP-AR1)	49.11 ± 5.72 ^lmn^	28.2 ± 6.12 ^u–x^	9.96 ± 1.37 ^yz^	3.82 ± 0.48 ^r^
*Lolium × hybridum* Hausskn. (LH-T)	58.88 ± 7.50 ^f–k^	36.28 ± 4.13 ^qr^	12.57 ± 1.76 ^xyz^	5.46 ± 0.39 ^o^
*X Festulolium braunii* (K. Richt.) (GR 5004)	48.61 ± 6.22 ^mno^	27.94 ± 6.24 ^wx^	9.66 ± 1.81 ^z^	3.41 ± 0.42 ^st^
*X Festulolium braunii* (K. Richt.) (GR 5003)	49.55 ± 7.47 ^l^	28.66 ± 5.73 ^tu^	11.38 ± 1.45 ^yz^	3.83 ± 0.55 ^r^
*X Festulolium* sp. (GR 5009)	48.5 ± 6.95 ^no^	27.91 ± 4.85 ^wx^	9.54 ± 1.49 ^z^	3.56 ± 0.70 ^s^
*X Festulolium* sp. (GR 1692)	48.16 ± 7.39 ^o^	27.71 ± 4.19 ^x^	9.48 ± 1.64 ^z^	3.23 ± 0.54 ^t^
*Paspalum dilatatum* Poir. (PASP 990)	59.05 ± 5.73 ^f–j^	37.13 ± 4.52 ^i–m^	13.07 ± 1.29 ^t–x^	6.01 ± 0.38 ^l^
LSD (5%)	0.865	0.539	0.274	0.234

**Table 5 plants-12-03358-t005:** Total phenolic and flavonoid contents.

Grass Genotypes	Total Phenolic Content (mg GAE/g DW)	Total Flavonoid Content (mg QE/g DW)
*X Festulolium* sp. (GR 5009)	7.76 ± 0.72 ^a^	0.52 ± 0.11 ^a^
*X Festulolium* sp. (GR 1692)	7.43 ± 0.54 ^b^	0.506 ± 0.17 ^b^
*X Festulolium braunii* (K. Richt.) GR 5004	7.13 ± 1.07 ^c^	0.503 ± 0.15 ^b^
*Dactylis glomerata* L. (D-M)	5.07 ± 0.29 ^d^	0.385 ± 0.04 ^c^
*Dactylis glomerata* L. (DG-P)	4.55 ± 0.52 ^e^	0.341 ± 0.02 ^d^
LSD (5%)	0.14	0.06

Means followed by the same letter in each column are not significantly different; according to the LSD test at the 5% level of probability.

**Table 6 plants-12-03358-t006:** Identified phenolic compounds of studied grass genotypes.

Grass Genotypes	GA	CGA	CA	PCA	FA	AP	VA	SyA	4HBA	Total
(μg g DW)										
GR 5009	23.8	103.3	388.1	179.9	153.1	148.1	190.6	280.6	84	1551.6
GR 1692	24.4	108.4	359.9	154.1	135.6	137.4	165.7	281.7	80.9	1448.4
GR 5004	64	105.9	327.4	182.8	102.6	109.6	166.4	285.4	83.4	1427.9
DG-M	34	63.5	130.1	133.9	95.8	115.1	129.3	248.2	76.9	1027.1
DG-P	29.1	55.2	82.5	123.9	86.5	148.9	127.5	281.8	nd	935.7

These compositions were detected and quantified by HPLC-MS; (nd): Not detected; GA: Gallic acid; CGA: Chlorogenic acid; CA: Caffeic acid; PCA: p-Cumaric acid; FA: Ferulic acid; AP: Apigenin acid; VA: Vanillic acid; SyA: Syringic acid; 4HBA:4-Hydroxybenzoic acid.

## Data Availability

Data are available from the corresponding authors upon request.

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
