# Peer review of "Allelopathic Effect of Aqueous Extracts of Grass Genotypes on Eruca Sativa L."

_plants, 2023, doi:10.3390/plants12193358_

Round 1

Reviewer 1 Report

Authors in a bioassay study examined the allelopathic effect of 50 grass species/genotypes on Eruca sativa as a recipient (test) species and identified important chemical compounds. Most important innovative content of the work is to study the intraspecific differences of the donor plants, which is poorly understood in allelopathic studies until now.

Research aims, literature review, description of the experiments, data evaluation (including statistical ones also) are well-developed, figures and tables help in undertstanding.

I take  only some minor remarks and questions:

l.83. please delete parentheses

Why Eruca sativa was chosen as a recipient (test species)? Please explain it in the manuscript. Under natural conditions this species can frequently interfere with  grasses? Do you have any data on the interaction (physical, chemical) between  E.sativa and studied grasses  under field conditions?

I think that bioassay (lab) studies are very important to evaluate allelopathic effects, but due to their  provocative characterization they are considered as the “first step” of allelopathic studies. I suggest to do pot and field experiments also in the future, because  the data of such experiments are closer to reality in open field conditions, and they can provide more reliable results how to use allelochemicals for the development of a potential bioherbicide. After minor corrections and additions mentioned above I suggest the manuscript for publication.

Author Response

REVIEWER#1

  • 83. please delete parentheses

Answer: The manuscript has been amended as suggested.

  • Why Eruca sativa was chosen as a recipient (test species)? Please explain it in the manuscript.

Answer: Although Eruca sativa L. has been known as a crop plant since long, but like some plants like rye and rapeseed, it can be a weed of cool-season crops like wheat. In Iran, severe infestation of this weed of wheat fields has been reported in some places like Isfahan and Karaj. Therefore, its biological control through the allelopathy of these grasses is important. For this reason,  Eruca sativa L. was investigated in this study. Because if we can control emerging weeds in the beginning, we will be more successful and their spread will be prevented. (Added to the manuscript).

(The importance of these contents is also mentioned in this reference.  Seed Germination and Seed Bank Dynamics of Eruca sativa (Brassicaceae): A Weed on the Northeastern Edge of Tibetan Plateau. Jia, C.-Z.; Wang, J.-J.; Chen, D.-L.; Hu, X.-W.  Front Plant Sci 2022, 13, doi:10.3389/fpls.2022.820925.)

  • Under natural conditions this species can frequently interfere with grasses? Do you have any data on the interaction (physical, chemical) between  sativa and studied grasses  under field conditions?

Answer:  Yes, it can interfere with crop growth in wheat fields and cold crops. But as far as we have checked, there is no information about the strength of competition and its allelopathy with grasses. For this purpose, this experiment was designed to check the allelopathic activity of cold season grasses on Eruca sative.

Reviewer 2 Report

The manuscript by Motalebnejad et al. evaluated the allelopathic activity of ten grass species and identified possible phenolic compounds in the genotypes of the species that have the highest allelopathic activity and inhibitory effect on Eruca sativa. I only have some minor suggestions.

L12: When mentioned a plant species for the first time in the paper, full Latin name should be used, and its English name also should be added. Please check it throughout the manuscript.

Line 120: The units for extract concentration should be added in Table 3.

Table 4: The specific concentrations for these extracts should be mentioned, so that these data can be compared between samples.

Section 2.2, 2.3, 2.4: the concentrations for the treatments in these data should be also added.

Line 158, 207: “E. sativa” should be italic.

Line 168, 171,321,326,333: Delete 'Significantly' or 'significant. Avoid the use of subjective terms that only adorn the text, but do not increase intellectually. Or “P < 0.5” can be added to replace them.

Table 5: According to the data, there may be no statistically significant difference at the 5% level of probability between GR5009, GR1692 and GR5004. Please check it again.

Line255: The discussion should be simplified. Some parts were already mentioned in the introduction and others are just a literature review. Focus alone or discuss your results.

Line 294: Same to L12.

Line 388-389: This sentence seems an arbitrary description. It can be deleted.

Line 407: Changed “ml” into “mL”, also in other parts.

Line 407: Changed “hours” into “h”, also in other parts.

Line 408, 416, 419: the units of Degree Celsius was wrong. It should be °C.

Author Response

REVIEWER#2

  • L12: When mentioned a plant species for the first time in the paper, full Latin name should be used, and its English name also should be added. Please check it throughout the manuscript.

Answer: The manuscript has been amended as suggested.

Line 120: The units for extract concentration should be added in Table 3.

Answer: The manuscript has been amended as suggested.

  • Table 4: The specific concentrations for these extracts should be mentioned, so that these data can be compared between samples.

Answer: Table 4 shows mean comparison of the seed germination indices  under the main effect of grass genotypes. It means that analysis of variance was done first and when it was observed that the effect of genotypes on germination indices was significant, comparison of means was investigated. Therefore, concentration should not be mentioned in this table because the purpose is to investigate the main effect of genotypes.

  • Line 294: Same to L12.

Answer: Its full name is given in line 291

  • Section 2.2, 2.3, 2.4: the concentrations for the treatments in these data should be also added.
  • Line 158, 207: “E. sativa” should be italic.

Answer: The manuscript has been amended as suggested.

  • Line 168, 171,321,326,333: Delete 'Significantly' or 'significant. Avoid the use of subjective terms that only adorn the text, but do not increase intellectually. Or “P < 0.5” can be added to replace them.

Answer: The manuscript has been amended as suggested.

  • Table 5: According to the data, there may be no statistically significant difference at the 5% level of probability between GR5009, GR1692 and GR5004. Please check it again.

Answer: Table 5 was checked again. Based on LSD values for total phenolic contents, there was a significant at the 5% level difference between these three genotypes. While based on LSD values for total flavonoid content, there was no significant at the 5% level difference between GR 1692 and GR 5004 genotypes. but there was a significant at the 5% level difference for total flavonoid content, between the GR 5009 genotype and the other two genotypes(GR 1692 and GR 5004 ).

  • Line255: The discussion should be simplified. Some parts were already mentioned in the introduction and others are just a literature review. Focus alone or discuss your results.

Answer: The discussion was completely revised. The sections that were mentioned in the introduction were removed and related discussions were included. Also, the coherence of the contents was checked and corrections were made.

  • Line 294: Same to L12.

Answer: The manuscript has been amended as suggested.

  • Line 388-389: This sentence seems an arbitrary description. It can be deleted.

Answer: This sentence was removed from the text.

  • Line 407: Changed “ml” into “mL”, also in other parts.

Answer: The manuscript has been amended as suggested.

  • Line 407: Changed “hours” into “h”, also in other parts.

Answer: The manuscript has been amended as suggested.

  • Line 408, 416, 419: the units of Degree Celsius was wrong. It should be °C.

Answer: The manuscript has been amended as suggested.

Reviewer 3 Report

This work presents the results of a laboratory experiment on the germination of Eruca sative under the influence of water extracts of 50 grass genotypes.

In general, the experiment is typical for a dose-response experiment, with five concentrations of water extracts and control.

I doubt preparing extract at the highest concentration: 1:1 and 1:0.75 of water: grass powder. How was it possible to receive such a high concentration considering the volume of powder concerning the volume of water? Please provide photos of the extracts.

Also, the grass genotypes you used have both cultivated grasses and weeds. What was the key to selecting the species? Why F. arundinacea is the most frequent species?

I don't find a justification for comparing cultivars of F. arundinacea with the other species. In my opinion, this approach diminishes possible interesting differentiation within different accessions of F. arundinacea, and the same relates to D. glomerata. I suggest comparing these species separately to extract differences within the species.

Discussion needs to be improved. It is too long now and not coherent. 

Round 2

Reviewer 3 Report

My questions have been addressed and corrections made. The paper is OK now.